# Nonlinear Distortion Cancellation using Predistorter in MIMO-GFDM Systems

**Ari Endang Jayati [1,2,]\*, Wirawan [1], Titiek Suryani [1] and Endroyono [1]**

1   Department of Electrical Engineering, Faculty of Electrical Technology, Institut Teknologi Sepuluh Nopember, Jalan Raya ITS, Keputih, Sukolilo, Surabaya 60111, Indonesia; wirawan@ee.its.ac.id (W.); titiks@ee.its.ac.id (T.S.); endroyono@ee.its.ac.id (E.)
2   Department of Electrical Engineering, Universitas Semarang, Semarang 50196, Indonesia
\*   Correspondence: ndhank80@gmail.com; Tel.: +62-858-6586-6740

**Abstract:** Generalized frequency division multiplexing (GFDM) with offset quadrature amplitude modulation (OQAM) is an alternative non-orthogonal modulation scheme for future generation wireless broadband systems. The nonlinearity of high power amplifiers (HPAs) has a very significant effect on the performance of GFDM systems. In this paper, we investigate the effects of nonlinear distortion on the multiple-input multiple-output (MIMO)-GFDM system when the signal is passed the HPA, which is modeled with amplitude and phase distortion. The effects of nonlinear distortion due to the HPA include amplitude distortion, phase distortion, and the spread of signal constellations. These effects also produce harmonic signals and intermodulation outside the frequency band which results in spectral spread. This will then reduce the performance of the MIMO-GFDM system. The contributions of this paper concern three key areas. Firstly, we investigate the effects of nonlinear distortion on the MIMO-GFDM system. We also derive the new closed-form expression bit error rate (BER) in MIMO-GFDM systems that use a memoryless HPA, which is modeled using the Saleh model when passed through the additive white Gaussian noise (AWGN) channel. This model was chosen because it is simple and has AM/AM and AM/PM curves. Secondly, we propose the application of techniques for the linearization of each HPA predistorter on the transmitter side of the MIMO-GFDM system separately. This predistorter is able to compensate for nonlinear distortion caused by the HPA without memory operating in the saturation region. The main contribution of this paper is to investigate the predistorter, which can linearize nonlinear distortion in MIMO-GFDM transmitters. The performance of the proposed scheme is evaluated in terms of spectrum analysis, PAPR analysis, a constellation diagram, and bit error rate (BER) analysis. The simulation results show that the proposed predistorter design succeeds in compensating for nonlinear distortions caused by the HPA for large input back-off (IBO) values.

**Keywords:** generalized frequency division multiplexing; predistorter; high power amplifier; multiple input multiple output; Saleh model

---

## 1. Introduction

The future generations of mobile networks, the fifth generation (5G), requires high data rates, flexibility, and also low latency. The 5G scenario includes increased system capacity, high quality of services (QoS), low power consumption, and low latency. Generalized frequency division multiplexing (GFDM) is one of the candidates of non-orthogonal waveforms that is currently being considered and evaluated by the 5GNOW team [1]. GFDM is able to overcome orthogonal frequency division multiplexing (OFDM) weaknesses, such as a high peak to average power ratio (PAPR). This is because OFDM signals are the sum of exponential terms limited to rectangular windows, while GFDM is the

sum of limited exponential terms with raised cosine (RC) filters. Given the assumption of having a normalized filter, the average power of the GFDM signal is certainly reduced by applying windowing, while remaining constant for OFDM. Therefore, PAPR is the same for the sum of exponential terms and OFDM signals, while it is statistically growing for GFDM [2]. Studies on PAPR in OFDM have been extensively carried out [3–6]. GFDM also produces low out-of-band (OOB) radiation due to the use of raised cosine pulse shaping to produce more efficient bandwidth. The cellular system and base station on 5G requires a new and faster processor, baseband, and radio frequency (RF) device. High power amplifiers (HPAs) are an important element of any communication system to communicate the signals to a significant distance [7]. HPAs are used to convert low power RF signals into large power signals to drive the transmitter [8]. This is because the need for coverage and long-distance transmission in wireless communication systems currently requires HPAs [9]. HPAs have three operating areas, the cutoff region, the linear region, and the saturation region. In general, the HPA will be operated in the linear region approaching the saturation region in order to increase power efficiency. When the HPA input power is close to the saturation point, it increases power efficiency and extends the battery life of the mobile transmitter. However, the HPA will produce nonlinear distortion if it is operated in this area [10]. Nonlinear distortion due to HPA causes several effects, both for the users themselves and for other users. The effects on nonlinear distortion include amplitude distortion, phase distortion, and the spread of signal constellations [11]. The effects of nonlinear distortion also produce harmonic and intermodulation signals outside the frequency band, which results in a transmitted spectral signal, which in turn interferes with the adjacent channel [10]. Outof-band rises because the nonlinear effects also cause interference between adjacent subcarriers [12]. New waveforms for the physical layer (PHY) and diverse requirements for 5G cellular systems are being studied, where one of the candidates is non-orthogonal GFDM. This new wave still uses high M-ary modulation [13]. Just like other multicarrier systems, GFDM also uses HPAs on the transmitter side. The use of HPAs on massive multiple-input multiple-output (MIMO) systems for 5G technology is the downlink at the base station system [14]. Like single-input-single-output (SISO), MIMO transceivers are also sensitive to nonlinear distortion due to HPAs. However, the problem becomes more complicated with the combined problem of HPA nonlinear distortion at the MIMO-GFDM with high M-ary modulation.

Recently, a study on HPA design for broadband and multiband has been proposed to support the base station transmitter on broadband communication systems [7]. Basically, research on RF design and implementation related to 5G systems with special emphasis on spectral contained waveform and small cell system scenarios has been conducted [15]. Furthermore, several studies on PAPR investigations, OOB radiation, and BER of the GFDM system if exposed to nonlinear distortion have been accomplished in recent years [16–18]. Generally, there are four types of techniques which may be used to overcome nonlinear distortion in the multicarrier system. These include decreasing input back-off (IBO) [19], linearization of the HPA [20], a PAPR reduction approach [12], and signal reconstruction on the receiver side by estimating nonlinear noise and eliminating it with an iterative algorithm to eliminate the probability of error [21]. However, there limited studies that examine the effects of nonlinear distortion and techniques to overcome distortion in GFDM. A study that discusses the performance of Long-Term evolution (LTE) MIMO downlink systems in various fading environments has been carried out [22]. Research on GFDM MIMO has been carried out on the time-reversal space-time coding technique, Alamouti space-time coding, and spatial multiplexing, among others [23–25]. The detection system on GFDM MIMO using expectation propagation and near-maximum likelihood [25] has also been investigated. Additionally, the following studies have also been carried out: the implementation of the $2 \times 2$ MIMO transceiver GFDM for the 5G [26], the effects of nonlinear distortion on MIMO GFDM [18], and nonlinearity research on the space-time coding (STC) MIMO system using the Saleh model [27], where the HPA of each branch is assumed to have the same nonlinear characteristics and the parameters are considered known at the receiver side. However, based on the central limit theorem and for large subcarrier values, nonlinear distortion can be considered as a complex Gaussian process [28]. For MIMO systems, each user operates with multiple antennas and therefore the identical

HPA assumptions are more realistic [11]. The techniques meant to overcome nonlinear distortion may be carried out on the transmitter or receiver sides. Predistorters are designed to model the inverse of a nonlinear HPA response and are often applied to the transmitter side for systems with a single transmitting antenna (SISO or SIMO system), though they may also be applied to MIMO systems [29] Two methods—DLA and ILA—may be compared to eliminate crosstalk on digital predistorters in the MIMO system where it is applied to each HPA separately [30]. On a large scale, antenna transmitter beam-forming has been investigated with the hybrid digital predistorter linearization method [31].

To date, no research has investigated the technique of overcoming nonlinear distortion in MIMO-GFDM systems. The contributions of this paper are as follows:

1. This study investigates the effect of HPA nonlinearity on MIMO-GFDM systems and derives the new signal-to-noise (SNR) ratios in the scenario of a nonlinear MIMO-GFDM system. The HPA is characterized using amplitude and phase distortion curves known as AM/AM and AM/PM curves and modeled using the famous Saleh model [32]. This is an empirical model used in the literature because of the simplicity of parameter extraction obtained using direct measurements.

2. The study applies a predistorter algorithm meant to linearize HPAs at MIMO-GFDM. This approach is called "local linearization" and is used to obtain a distortion-free signal at the output of each HPA. For simplicity, it is assumed each of the HPAs in all branches of the sender have the same nonlinear characteristics. Furthermore, to simplify the model, the HPAs are presumed to be modeled using the Saleh model. The idea of modifying the MIMO-GFDM system to obtain distortion-free signals is investigated [13,33].

3. The main contribution is to investigate predistorter, which can linearize nonlinear distortion in MIMO-GFDM transmitters. The predistorter proposal with local linearization in GFDM MIMO uses a point to point scenario. The results are supported by Matlab simulation of the MIMO-GFDM system in the Additive White Gaussian noise (AWGN) channel. This application requires changes on the transmitter side (i.e., at the base station on cellular networks) and not on the receiver.

The remaining parts of this report are organized as follows: Section 2 presents the system model, HPA characteristic, and the proposed algorithm; Section 3 includes the results and discussion; and Section 4 presents the conclusions.

## 2. Methods

### 2.1. System Model

GFDM system with nonlinear HPA assuming the system uses Offset QAM mapping, as shown in Figure 1. After passing through the mapper, the data were modulated using GFDM. In a GFDM modulator, a serial OQAM symbol stream was converted into parallel data, which were broken into multiple blocks with size $K \times M$. The $K$ denotes the number of subcarriers, while the $M$ sub-symbol is data transmitted on the k-th. Each sub-symbol results from the decomposition, upsampled by factor $N$, to be converted into an impulse signal sequence.

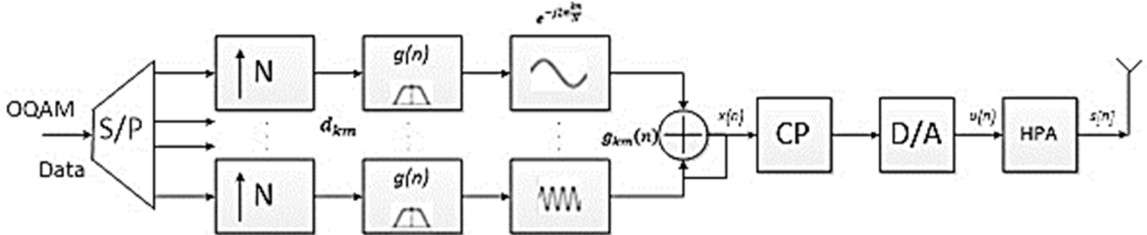

**Figure 1.** Generalized frequency division multiplexing (GFDM) system with high power amplifier (HPA).

The resulting signal was shifted in a circle and transmitted in the form of pulses in the following equation [34]:

$$g_{k,m}(n) = g\left[\left(n - \frac{mK}{2}\right) \bmod KM\right] e^{\frac{j2\pi k}{K}\left(n - \frac{Lp-1}{2}\right)}, \tag{1}$$

where $n = 0, 1, \ldots, KM - 1$, and $Lp$ is the length of the prototype filter, $g_{k,m}(n)$ is a prototype filter $g(n)$ with the shifted version in time and frequency, $k$ is the subcarrier index, and $m$ is for the sub-symbol index. The prototype filter above consists of two components, namely $\left[\left(n - \frac{mK}{2}\right) \bmod KM\right]$ and complex exponential numbers. The first component is a circular filter that functions as a time slider to distinguish between one symbol and the other symbols based on the timeslot. Meanwhile, the second component functions as a signal shifter in the frequency domain.

GFDM output signal $x[n]$ was written as follows [35]:

$$x[n] = \sum_{k=0}^{K-1} \sum_{m=0}^{M-1} u_{k,m} g_{k,m}[n] \tag{2}$$

or in the form of a matrix [35]:

$$x = Ad, \tag{3}$$

where: $x = (x[n])^T$, contains samples $x[n]$, $d = \text{vec}(D^T)$, where $D$ is a matrix containing complex value data symbols arranged in the $K \times M$ dimension matrix and $A$ is a matrix containing $g_{k,m} = (g_{k,m}[n])^T$, as shown in the following equation:

$$A = \left[g_{0,0} \cdots g_{K-1,0} \cdots g_{0,1} \cdots g_{K-1,M-1}\right]. \tag{4}$$

The data were modulated and then converted into serial data in the block parallel-to-serial or P/S. Added cyclic prefix and HPA then passed the AWGN channel. The cyclic prefix is added to GFDM to maintain the circular structure of the transmission signal and to allow the frequency domain even distribution to the receiver after a multi-lane effect is applied in the channel [36]. The insertion of a cyclic prefix is in each block of GFDM instead of in each symbol, which increases the spectral efficiency [35]. In the MIMO-GFDM system, cyclic prefix (CP) is taken from the final copy of each of the GFDM symbols, and then it is placed at the beginning of the frame. The number of symbols in this study is $K \times M$, which is 120, so the amount of CP used is $0.25 \times K = 0.25 \times 8 = 2$.

For simplicity, all HPAs in each branch were assumed to have the same nonlinear characteristics. Furthermore, to simplify the model, the HPA was presumed to be memoryless. They could be modeled using AM/AM and AM/PM conversion and written as follows [32]:

$$A[r(n)] = \frac{\alpha_\alpha r(n)}{1 + \beta_\alpha r^2(n)}, \tag{5}$$

continuing with the following AM/PM characteristic functions:

$$\Phi[r(n)] = \frac{\alpha_\varphi r^2(n)}{1 + \beta_\varphi r^2(n)}, \tag{6}$$

where $\alpha_\alpha$, $\beta_\alpha$, $\alpha_\varphi$, $\beta_\varphi$ are the parameters of the Saleh model [32], while $r(n)$ is the magnitude of the input signal. The functions $A[r(n)]$ and $\Phi[r(n)]$ are two-parameter formulas such as Equations (5) and (6). $A[r(n)]$ is a function of an odd number of r, which represents the AM-to-AM conversion, and $\Phi[r(n)]$ is a function of an even number of r, which represents the AM-to-PM conversion. If the value of r is very large, $A[r(n)]$ is proportional to $1/r$, and $\Phi[r(n)]$ approaches constant.

### 2.2. GFDM Predistorter

This research focused on the HPA linearization technique with predistorter on the transmitter side in the BS for the 5G downlink system. The predistorter was chosen because it is a simple HPA linearization method, limiting the regrowth spectrum and eliminating the effect of amplitude distortion. It was designed to model the inverse of a nonlinear HPA response. The device was then placed before the non-linear HPA to obtain a linear output signal, as shown in Figure 2.

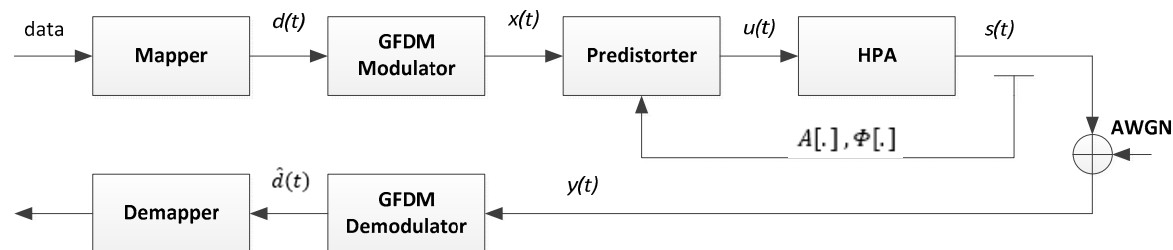

**Figure 2.** GFDM system with predistorter.

With a HPA and predistorter, the output of the GFDM modulator may be written as in the following equation [37]:

$$x(n) = \rho(n)e^{j\Phi(n)}, \tag{7}$$

where $\rho(n)$ is the amplitude of the GFDM modulator output signal and $\phi(t)$ is the signal phase of the GFDM modulator output. The output of the predistorter is formulated as follows [38]:

$$u(n) = r(n)e^{j\theta(n)}, \tag{8}$$

where $r(n)$ is the amplitude of the HPA input and $\theta(n)$ is the phase of the HPA input.

To compensate for nonlinearity, it must satisfy the equation: $x(n) = s(n)$, such that:

$$r(n) = A^{-1}[\rho(n)],$$
$$\theta(n) = \phi(n) - \Phi[r(n)], \tag{9}$$

where $A\,[.]$ and $\Phi\,[.]$ are the AM/AM and AM/PM characteristic functions of HPA local linearization and should cancel the distortion in each HPA, so that the signal at the transmitter at each branch is in a linear area. The inverse $A\,[.]$ function will exist if $\rho(n) \geq \sqrt{\frac{\alpha_\alpha{}^2}{4\beta_\alpha}}$, so the predistorter input must be limited not to exceed $\sqrt{\frac{\alpha_\alpha{}^2}{4\beta_\alpha}}$. The predistorter output can be written as follows [39]:

$$r(n) = \begin{cases} \frac{\alpha_\alpha - \sqrt{\alpha_\alpha{}^2 - 4\beta_\alpha \rho(n)^2}}{2\beta_\alpha \rho(n)}, & 0 < \rho(n) < \sqrt{\frac{\alpha_\alpha{}^2}{4\beta_\alpha}} \\ A^{-1}\left(\sqrt{\frac{\alpha_\alpha{}^2}{4\beta_\alpha}}\right) = \frac{1}{\sqrt{\beta_\alpha}}, & \rho(n) \geq \sqrt{\frac{\alpha_\alpha{}^2}{4\beta_\alpha}} \end{cases},$$

$$\theta(n) = \phi(n) - \frac{\alpha_\phi r^2(n)}{1 + \beta_\phi r^2(n)}. \tag{10}$$

For ideal predistortion (assuming the Saleh model for HPA is exact), the input phase of the signal does not change, but the amplitude does. Therefore, the characteristic functions of AM/AM and AM/PM are

$$A(\rho(n)) = \begin{cases} \rho(n), & 0 < \rho(n) < \sqrt{\frac{\alpha_\alpha{}^2}{4\beta_\alpha}} \\ \sqrt{\frac{\alpha_\alpha{}^2}{4\beta_\alpha}}, & \rho(n) > \sqrt{\frac{\alpha_\alpha{}^2}{4\beta_\alpha}} \end{cases}$$

$$\Phi(\rho(n)) = 0 \tag{11}$$

with HPA output as

$$s(n) = R(n)e^{j\psi(n)}, \tag{12}$$

where $R(n) = A[r(n)]$ is the output amplitude and $\psi(n) = \theta(n) + \Phi[r(n)]$ are phase output HPA.

### 2.3. Proposed Predistorter for MIMO-GFDM Systems

The classic approach for the MIMO-GFDM systems with nonlinear HPA is that each HPA is linearized separately by the local predistorter, can be seen in Figure 3. The purpose of this local predistorter was to reduce the nonlinear distortion of each HPA. In this MIMO system, a point to point scenario was assumed. The receiver collects the signals from the receiving antenna and processes them to decode the data stream. All HPAs were assumed to have the same nonlinear characteristics in each branch and were following the memoryless Saleh model.

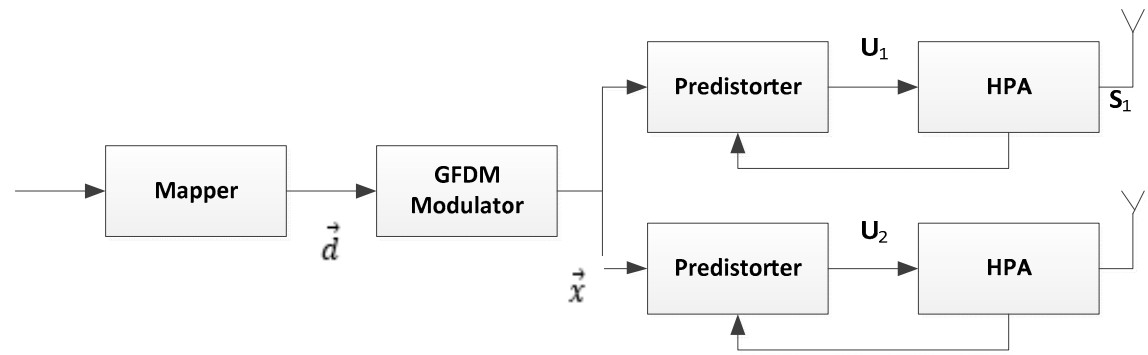

**Figure 3.** MIMO-GFDM system with predistorter.

A complex envelope from the HPA input from each branch was formulated as follows:

$$x_k(n) = \rho_k e^{j\theta_k}, \tag{13}$$

where $\rho_k$ is the amplitude of $x_k$, $\theta_k$ is the phase of the input $x_k$, and $k$ is the number of antennas.

The complex envelope of the HPA output can be modeled as follows:

$$\begin{aligned} s_k(n) = g[x_k(n)] \quad &= A(\rho_k)e^{j(\theta_k + P(\rho_k))}, \\ &= S(\rho_k)e^{j\theta_k}, \end{aligned} \tag{14}$$

where $g[.]$ is the transfer function of HPA, $A(.)$ And $P(.)$ is the conversion of AM/AM and AM/PM to HPA, and $S(\rho_k))$ is the amplitude of the HPA output. For the nonlinear PA model, the Saleh model such as in Equations (5) and (6) was used.

The operating point on the HPA is normally identified by a 'back-off' for the validation of the theoretical results, defined as:

$$IBO = 10log_{10}(\frac{A_0^2}{P_0}), \tag{15}$$

where $A_0$ is the saturation of the input ampitude HPA and $P_0$ is the average input power. From Bussgang's theorem and by extending complex Gaussian processes, the output of the HPA may be expressed as:

$$s_k(n) = K_0 x_k(n) + d_k(n), \tag{16}$$

where $K_0$ is the attenuation coefficient and $d_k(n)$ is a nonlinear distortion not correlated with the input signal $x_k(n)$. The $K_0$ value given by [40] with

$$K_0 = \frac{1}{2}E\left[S'(\rho_k) + \frac{S(\rho_k)}{\rho_k}\right], \tag{17}$$

where $S'(\rho_k)$ is denoted as a derivative of the $S(\rho_k)$. The variance value of nonlinear distortion $d_k(n)$ is as follows [41]:

$$
\begin{aligned}
\sigma_d^2 &= E[|d_k(n)|_2], \\
&= E[|z_k(n)|_2] - |K_0|_2 E[|x_k(n)|_2], \\
&= E[|S(\rho_k)|_2] - |K_0|_2 E[\rho_k^2].
\end{aligned}
\tag{18}
$$

The signal received at each $y_m^{NL}(n)$ antenna is formed by the independent superposition of the signal. This is related to the $N_t$ antenna that shares the same space frequency resources. This signal is related to the $N_t$ antenna which shares the same frequency space. The received signal was exposed to Gaussian noise on the array element as follows:

$$
y_m^{NL}(n) = \sum_{k=1}^{N_t} H_{m,k}(n)(K_0 x_k(n) + d_k(n)) + n_m(n),
\tag{19}
$$

where $n_m(n)$ is additive white Gaussian noise (AWGN) and $H_{m,k}(n)$ is a circular time domain channel matrix $Nc \times Nc$ at time $n$ formed from the $h_{m,k}(n)$ channel response vectors for the link between the $k$ transmitting and $m$ receiving antennas. The channel matrix becomes a circular matrix because of the application of a cyclic prefix before transmission. In addition, the decomposition of the eigenvectors leads to the diagonalization of the channel matrix [42]. While at the receiver side, using time domain equalization techniques offers better performance compared to frequency domain [43].

The received signal in the frequency domain may be written as follows:

$$
Y_m^{NL}(n) = V y_m^{NL}(n),
\tag{20}
$$

where $V$ is the matrix of the Fast Fourier Transform (FFT) $V \in \mathbb{C}^{Nc \times Nc}$ with the $Nc$ being the number of subcarriers. The signals received in the form of vectors for each antenna at the $i$-th subcarrier may be written as follows [41]:

$$
Y^{NL}(n,i) = K_0 \mathcal{H}(n,i)X(n,i) + \mathcal{H}(n,i)D(n,i) + \mathcal{N}(n,i),
\tag{21}
$$

where $\mathcal{H}(n,i)$ is the transfer matrix of channels in the frequency domain, $X(n,i) = [X_1(n,i) \dots, X_{Nt}(n,i)]^T$ is the vector containing the signal transmitted by each antenna, and $\mathcal{N}(n,i)$ is the Gaussian noise obtained using the FFT process of the thermal Gaussian noise with zero mean and $\sigma^2$ variance.

The steps of the predistorter on the MIMO-GFDM system are summarized in the Algorithm 1 as follows:

---

**Algorithm 1** Modelling of Predistorter for MIMO-GFDM Systems

---

| | | |
|---|---|---|
| 1: | $\mathbf{d} \leftarrow \mathbf{b}$ | ▷ mapping 16 OQAM |
| 2: | $\mathbf{A} \leftarrow \mathbf{g}$ | ▷ pulse shaping RRC |
| 3: | $\mathbf{x} \leftarrow \mathbf{Ad}$ | ▷ output GFDM |
| 4: | **for k = 1 : K do** | ▷ per Tx branch |
| 5: | $x_k(n) \leftarrow \rho(n)e^{j\phi(n)}$ | ▷ input predistorter |
| 6: | $r_k(n) \leftarrow A^{-1}[\rho(n)]$ | ▷ amplitude predistorter |
| 7: | $\theta_k(n) \leftarrow \phi(t) - \Phi[r(t)]$ | ▷ phase of predistorter |
| 8: | $u_k(n) \leftarrow r_k(n)e^{j\theta_k(n)}$ | ▷ output predistorter |
| 9: | $R_k(n) \leftarrow A[r_k(n)]$ | ▷ amplitude HPA |
| 10: | $\psi(n) \leftarrow \theta_k(n) + \phi[r_k(n)]$ | ▷ phasa HPA |
| 11: | $s_k(n) \leftarrow R_k(n)e^{j\psi(n)}$ | ▷ output HPA |
| 12: | **end for** | |

---

*2.4. HPA Characteristic*

The nonlinear distortion model used in this study was the Saleh model because it is quasi-memoryless with high accuracy and appropriate complexity. It is called a quasi-model because it has four parameters. This is also called memoryless PA if it has flat frequency response characteristics in all ranges or narrow band communication systems [10]. Memoryless power amplifiers have AM/AM characteristic functions (amplitude to amplitude) and AM/PM (amplitude to phase), which only depend on the current input signal value with Saleh model parameter values of $\alpha_\alpha = 2.1587$, $\beta_\alpha = 1.1517$, $\alpha_\phi = 4.033$, and $\beta_\phi = 9.1040$, taken from experimental data [32].

The first curve is the AM/AM function, which is the ratio of input and output amplitudes of the HPA. The second one is the AM/PM function, which is the ratio of input amplitude to the output phase of the HPA. The AM/AM characteristics were obtained from the measurement shown in Figure 4 by plotting the output power as a function of input power. From the characteristics of AM/AM, it is evident a maximum output power of 7 dB was not achieved. Similarly, AM/PM characteristics obtained by mapping the output phase shift as a function of input power are shown in Figure 5. AM/PM characteristics have been normalized to produce a zero degree phase shift at −10 dB.

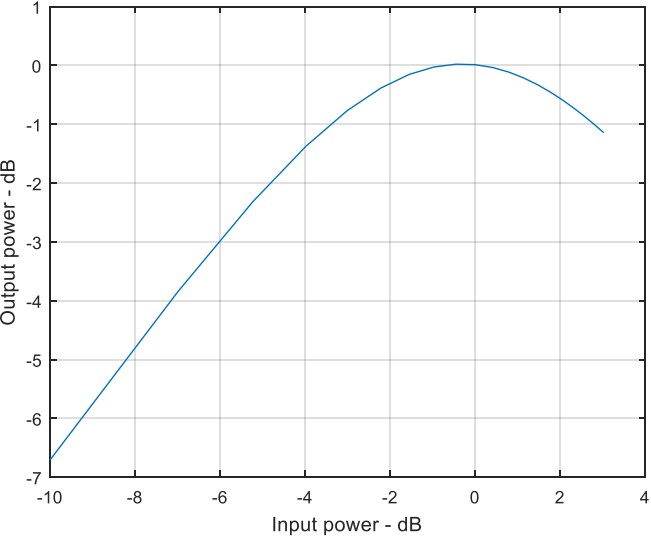

**Figure 4.** AM/AM characteristic curve of the Saleh model.

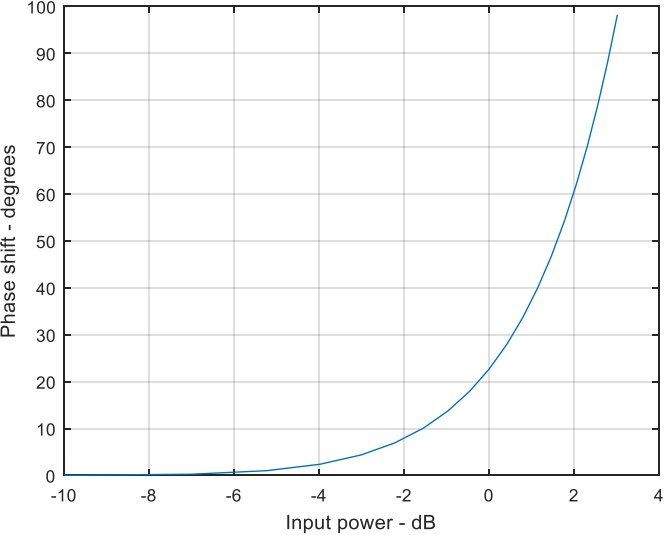

**Figure 5.** AM/PM characteristic curve of the Saleh model.

*2.5. Performance of MIMO-GFDM*

In this section, the performance of GFDM MIMO for symbol error rate (SER) forms was evaluated. MIMO-GFDM system simulation is performed with predistorters using Matlab according to Equation (10). The system used OQAM on the AWGN channel. SER for the GFDM/OQAM system through the AWGN channel was as follows [34]:

$$p_{AWGN}(e) = 2\left(\frac{k-1}{k}\right)erfc\left(\sqrt{\gamma}\right) + -\left(\frac{k-1}{k}\right)erfc^2\left(\sqrt{\gamma}\right). \tag{22}$$

SNR system GFDM/OQAM has the following equation:

$$\gamma = \frac{3R_T}{2(2^\mu - 1)} \cdot \frac{N_s E_s}{\zeta N_o} \tag{23}$$

and

$$R_T = \frac{KM}{KM + N_{CP} + N_{CS}}. \tag{24}$$

$\mu$ is the number of bits per symbol QAM, $p = \sqrt{2^\mu}$, $N_{CP}$ is the length of the cyclic prefix and $N_{CS}$ cyclic suffix, $K$ and $M$ represent the number of subcarriers and subsymbol, $E_s$ is the average energy per symbol, and $N_o$ is the noise power density. Factor $N_s$ has a value of 2 for OQAM and 1 for QAM.

Meanwhile, the SNR for GFDM/OQAM systems with HPAs from Equation (19) is:

$$\gamma = \frac{|K_o|^2 |H|^2 \sigma_x^2}{|H|^2 \sigma_d^2 + \sigma_n^2} \tag{25}$$

## 3. Results and Discussion

*3.1. Predistorter in MIMO-GFDM*

The application of a mitigation methods predistorter for nonlinear distortion due to the HPA on MIMO GFDM was investigated. The covered analysis included the system performance, which was BER, constellation diagram, and signal spectrum before and after given a nonlinear distortion. The GFDM simulation was carried out with the following parameters: $K = 8$ subcarriers, $M = 15$ subsymbol, mapping using 16 OQAM modulation, and the root raised cosine with a roll-of-factor 0.3, as shown in Table 1.

**Table 1.** Simulation parameters.

| Parameter | Notation | GFDM |
|---|---|---|
| sampling Frequency | $f_s$ | 16 |
| Subcarrier | K | 8 |
| Symbols per block | M | 15 |
| Samples per symbol | N | 10 |
| Pulse Shaping | g | RRC |
| Roll-of-factor | $\alpha$ | 0.3 |
| Alpha_a | $\alpha_\alpha$ | 2.1587 |
| Alpha_phi | $\alpha_\varphi$ | 4.0033 |
| Beta_a | $\beta_\alpha$ | 1.1517 |
| Beta_phi | $\beta_\varphi$ | 9104 |
| Mapping | | 16 OQAM |

With a predistorter, the AM/AM curve became linear as shown in Figure 6. This occurred because the predistorter inversed the nonlinear distortion, showing it was the simplest technique for the linearization of HPAs.

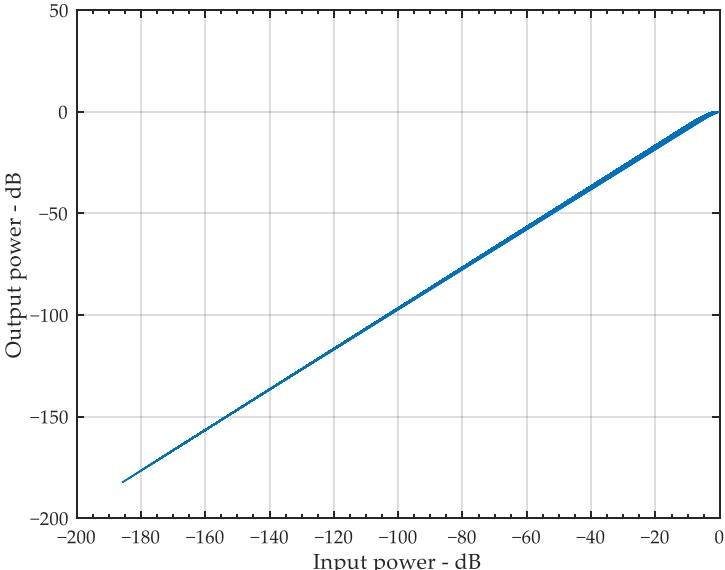

**Figure 6.** AM/AM characteristic curve of predistorter.

*3.2. Spectrum Analysis*

The first analysis was GFDM spectrum scrutiny, which was carried out with and without a predistorter as shown in Figure 7. The power spectral density (PSD) of complex signals is no longer symmetrical around *Fs*/2 even though it is still of real value. The *x* axis is the sampling frequency *Fs* and the *y* axis is the normalized PSD. The PSD of GFDM after being given predistorter resulted in low OOB while the inband was better. The aim of the predistorter is to modify the signal on the transmitter, so the nonlinear distortion seen in the PA output decreases [30]. Predistorters can be implemented as inverses directly from the characteristics of the HPA.

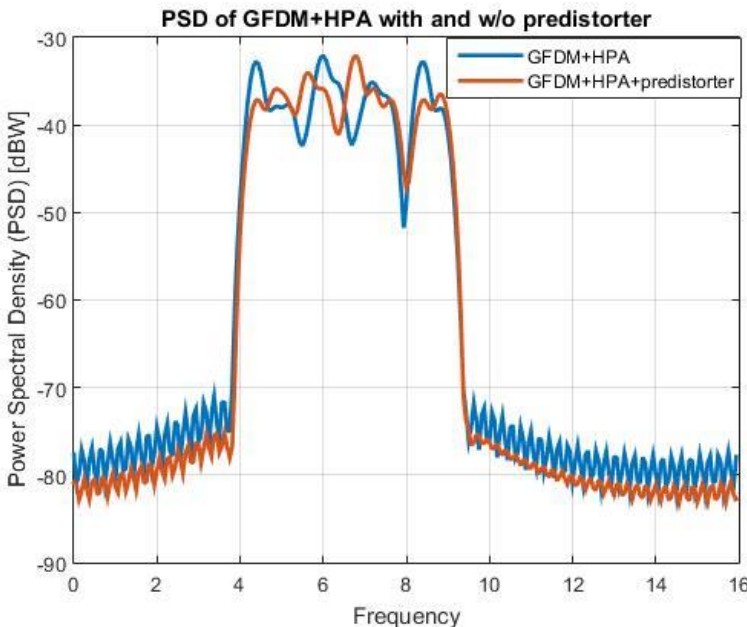

**Figure 7.** PSD SISO GFDM + HPA with and without predistorter.

The signal spectrum in GFDM was analyzed with the predistorter varied by IBO as shown in Figure 8. It was evident the IBO was getting smaller, the nonlinear distortion was enlarging, the OOB was increasing, and the inband was rippling.

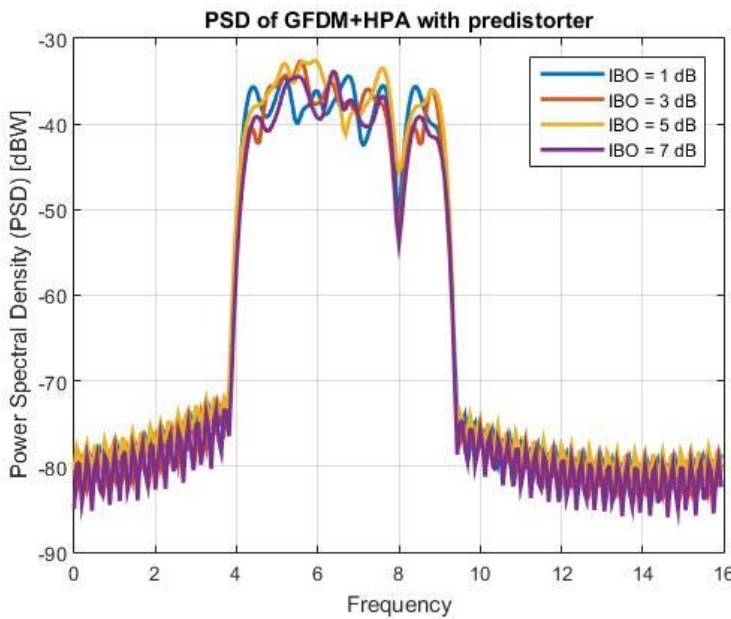

**Figure 8.** Comparison of PSD GFDM+predistorter with input back-off (IBO) variations.

Furthermore, the application of predistorter on the effect of the MIMO-GFDM signal spectrum was as shown in Figure 9. The MIMO-GFDM signal spectrum with predistorter become better than the system without predistorters. This approach has several advantages, such as the reduction of the out-of-band distortion. Nonetheless, the estimated PA memory is needed at the transmitter.

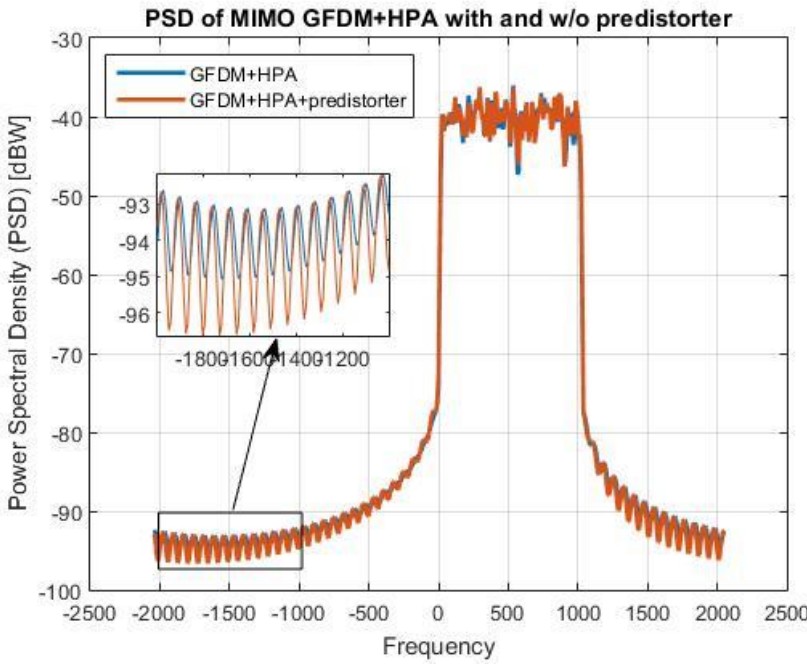

**Figure 9.** MIMO PSD GFDM with and without predistorter.

PA operating points need to be carefully selected to meet the spectral mask and should be optimized to get maximum power efficiency. These two constraints—power efficiency and adjacent channel interference (ACI)—are the key factors in a PA design.

*3.3. PAPR Analysis*

The second analysis involved the calculation of PAPR GFDM before and after the predistorter was given. PAPR occurs because in a multicarrier system it consists of a number of subcarriers modulated independently. Linearity requirements can be met by moving the PA far below the saturation point, but this causes poor power efficiency at the mobile transmitter. PA will be more power efficient if it operates close to the saturation region. Similarly, if the PA gets closer to the saturation region, the amount of nonlinear distortion is greater, which will reduce the system performance. Consequently, the nonlinear distortion produces harmonic and intermodulation signals outside the desired frequency band, which is disturbing for other users, especially in the adjacent frequency band. Therefore, interference must be reduced for both systems to operate satisfactorily.

The PAPR value is greater than 5 dB because GFDM, like other multicarrier systems, has a high PAPR value. The simulation results show the PAPR GFDM before and after the predistorter was given, as shown in Figure 10. For thecomplementary cumulative distributive function (CCDF) curve presented, the x-axis is the PAPR (dB), while the y-axis is the probability (X ≤ x). By applying the predistorter, the PAPR of GFDM dropped from 6.8 to 5.2 with a probability of $10^{-2}$. The visible performance is very good, which should be around the value of 5 dB for it to be applied in an environment with thousands of users.

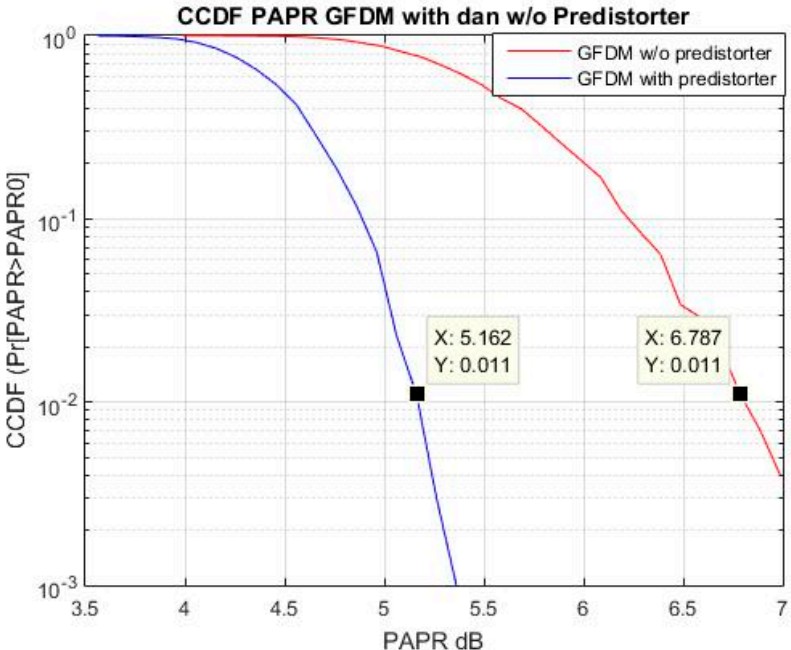

**Figure 10.** Comparison of PAPR GFDM with and without predistorter (IBO = 3dB).

Figure 11 is the result of CCDF display on a GFDM system with IBO variations. From the graph, if the IBO value gets bigger, the CCDF from the GFDM PAPR will decreases. Nonlinear distortion gets smaller if the IBO value gets higher. As a result, the HPA works in a linear area.

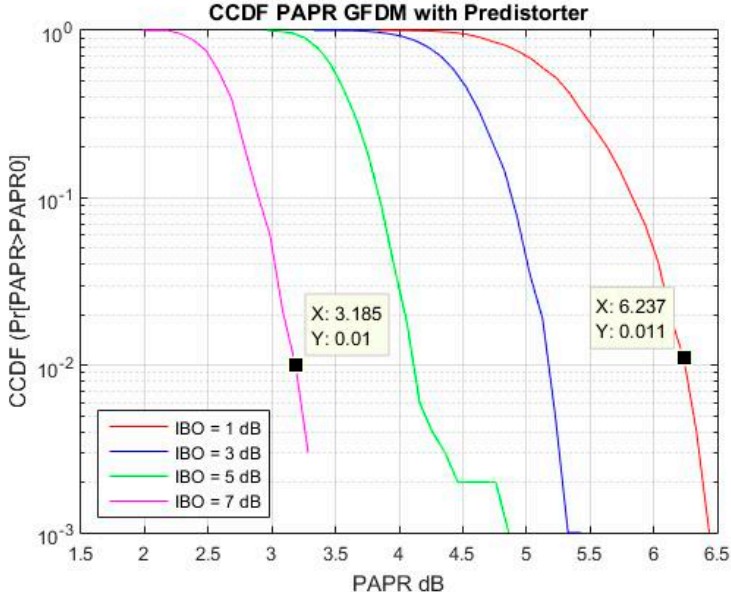

**Figure 11.** Comparison of PAPR GFDM+predistorter with IBO variations.

### 3.4. Constellation Diagram

The next effect analyzed was the constellation diagram after the GFDM was given a MIMO nonlinear distortion, as shown in Figure 12. Phase distortion occurs when an amplifier does not delay all frequency components by the same amount. The output phase distortion has a logarithmic shape instead of linearly increasing, as in the Saleh model. This means that the output phase shift is almost constant at high input amplitude values [9]. Phase distortion causes the constellation of signals on the receiving side to be no longer at the original point. The constellation diagram of the signal on the receiving side was no longer at the original point, called the "warping" effect, and there was also a spread of signal constellations in small groups, called effects "clustering". This will affect the decision point area on the receiver side. As a result, the received signal was difficult to detect. If detected, the received signal had an error, resulting in a decrease in performance. The constellation diagram for MIMO-GFDM exposed to nonlinear distortion experienced an enlargement of the decision point area. If the IBO gets smaller, the nonlinear distortion gets bigger and the decision point in the constellation also enlarges.

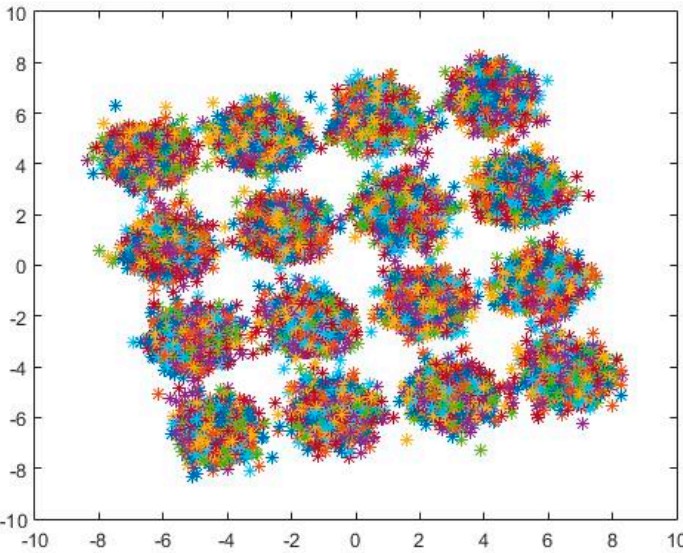

**Figure 12.** GFDM constellation without predistorter.

From the results of the investigation in Figure 13, after the MIMO-GFDM was given a predistorter the constellation returned to its original point in order not to interfere with the detection process on the receiving side. The signal is again not in a distorted phase and amplitude. From the constellation picture, there was a division of the decision region so that the detection process follows the changes. This could be perceived as an error vector getting better after being given a predistorter. The error vector magnitude (EVM) for GFDM without predistorter was -4.97 dB, and after the GFDM system the predistorter decreased to -9.0 dB.

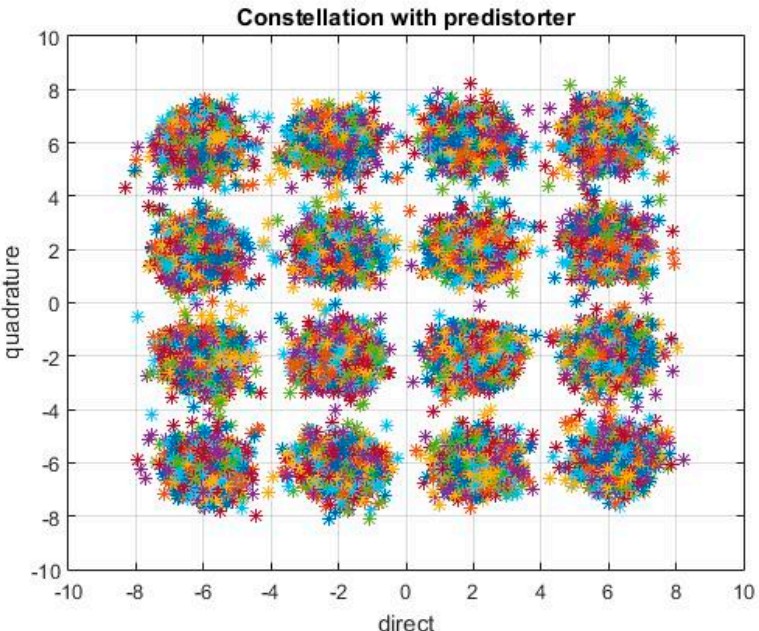

**Figure 13.** GFDM constellation with predistorter.

### 3.5. BER Performance

The last analysis is BER study. If GFDM MIMO is given a nonlinear distortion, the performance of BER will decrease, as shown in Figure 14. The proposed method has been compared with OFDM. The effects of nonlinear distortion in OFDM and GFDM have been investigated by [33]. The effect of nonlinear distortion in OFDM will increase PAPR, increase OOB, and decrease SER. Meanwhile, the effect of nonlinear distortion on the GFDM system increases the OOB signal and decreases the SER significantly [13]. Then, nonlinear distortion on the MIMO-GFDM system caused the spread of the input signal diagram and affected the decision point area on the receiving side. As a result, the received signal was harder to detect correctly. So that the system performance decreases because there are errors in the information received. The simulation results show the performance of the MIMO GFDM with predistorter was better than the MIMO GFDM without predistorter. In the next work, Alamouti space-time coding will be added to the MIMO-GFDM system in order to decrease the needed signal-to-noise ratio.

Theoretical BER was calculated by the Euclidean distance between the two closest signal points in the constellation diagram, which reduces the probability of errors. The probability of BER increased with the modulation size. Figure 15 shows the result of an investigation of the influence of IBO on the GFDM MIMO system if given a nonlinear distortion. It can be seen in the graph that if the IBO increases, the BER value in the GFDM MIMO system gets smaller. This is because at a larger IBO, the nonlinear distortion is smaller.

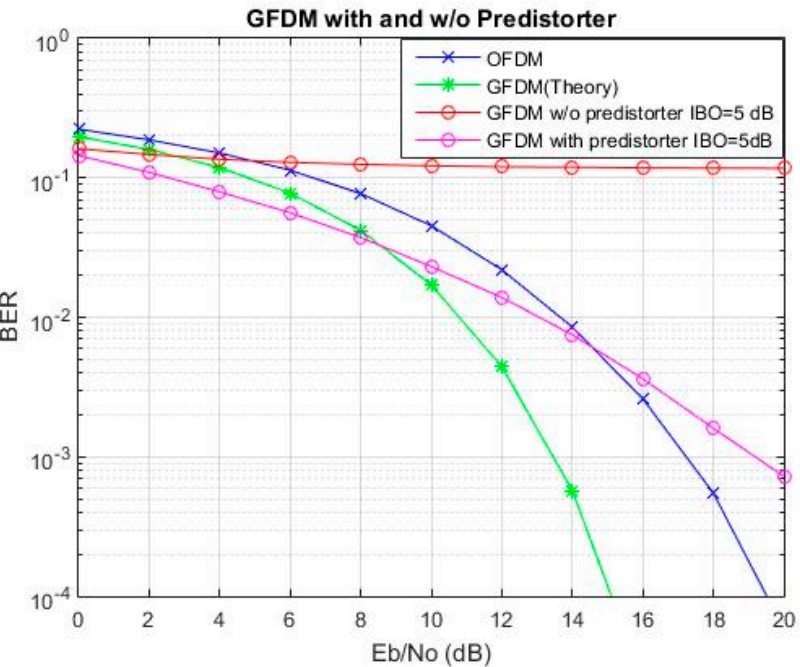

**Figure 14.** Comparison of GFDM with and without predistorter.

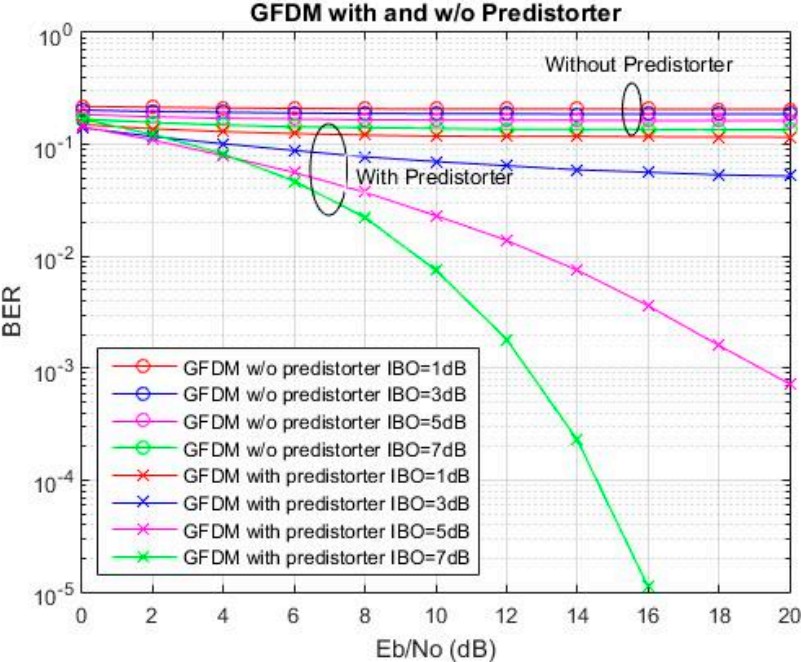

**Figure 15.** Bit error rate (BER) of GFDM+predistorter with IBO variations.

The comparison between BER and IBO for the GFDM system with HPA with and without predistorters is shown in Figure 16. The negative sign on IBO is used to simplify the analysis of simulation results. Looking at the IBO 6 dB, the GFDM with HPA system has the largest BER of 0.1668, while the system GFDM with predistorter has a lesser BER, i.e., 0.0001. Therefore, it can be concluded that the predistorters linearize nonlinear distortions successfully due to the HPA for larger IBO values. Future studies should focus on small IBO values to have a smaller BER value.

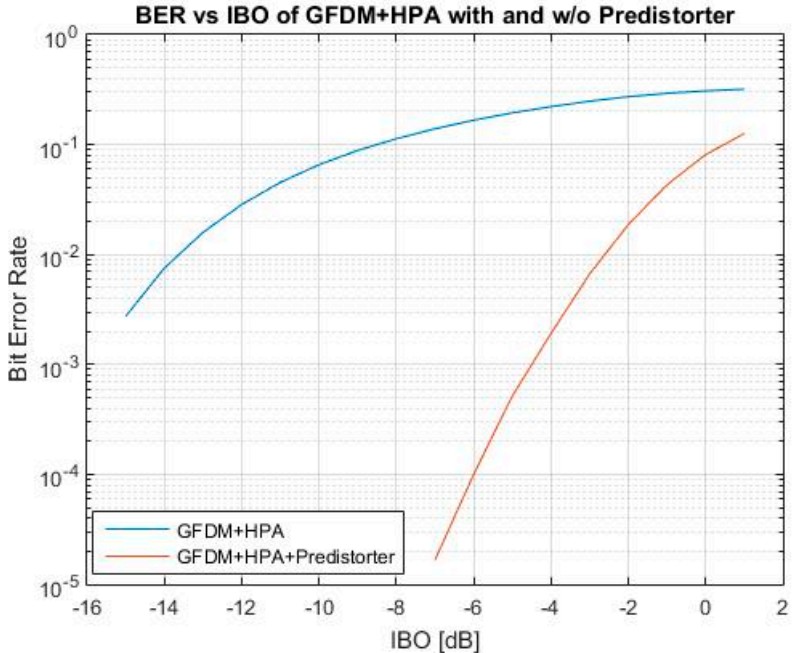

**Figure 16.** BER vs. IBO GFDM with and without predistorter.

## 4. Conclusions

This study investigated and analyzed the effect of HPA nonlinear distortion GFDM MIMO on the shape of the spectrum, the PAPR, and the BER in the AWGN channel. This research has derived the closed-form signal-to-noise ratio (SNR) in a nonlinear MIMO-GFDM system scenario. Furthermore, it applied predistorter techniques to mitigate distortion effects of the HPA nonlinear MIMO GFDM on the sender side.

The effectiveness of the proposed predistorter was illustrated by the simulation results which showed the performance comparison of MIMO GFDM with the HPA without predistorter. The results showed that application of predistorter at MIMO-GFDM can reduce the effects of HPA nonlinear distortion. The system with predistorter was shown to have better spectrum parameters, good PAPR, and lower BER values. The predistorter compensated for serious nonlinear distortion caused by the HPA memory which operated in the saturation region. It is clear that the transmitter processing method reduces the effect of nonlinear distortion.

This study only overcame nonlinear distortions in MIMO-GFDM system transmitters. In MIMO systems, interference is not only due to nonlinear distortion, but also to crosstalk, as a result of the interference between different signal paths. Therefore, a compensation technique for the joint effect of nonlinear distortion and crosstalk in MIMO-GFDM systems should be considered in future studies. In the next work, Alamouti space-time coding will be added to the MIMO-GFDM system.

**Author Contributions:** Conceptualization, A.E.J., W., and T.S.; methodology, A.E.J., W., and T.S.; software, A.E.J. and T.S.; validation, A.E.J. and W.; formal analysis, A.E.J., W., T.S., and E.; investigation, A.E.J., W., T.S., and E.; resources, A.E.J.; data curation, A.E.J.; writing—original draft preparation, A.E.J.; writing—review and editing, A.E.J., W., T.S., and E.; visualization, A.E.J., W., T.S., and E.; supervision, W., T.S., and E.; project administration, A.E.J., W., T.S., and E.; funding acquisition, A.E.J., W., T.S., and E.

**Funding:** This research was funded by the Ministry of Research, Technology, and Higher Education throughthe scholarship program BUDI-DN by the LPDP of the Ministry of Finance of the Republic of Indonesia to Ari Endang Jayati.

**Acknowledgments:** The authors acknowledge the support for this research from LPPM of Institut Teknologi Sepuluh Nopember Surabaya, Indonesia.

**Conflicts of Interest:** The authors declare no conflicts of interest.

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
