# Peer review of "Nonlinear Distortion Cancellation using Predistorter in MIMO-GFDM Systems"

_electronics, doi:10.3390/electronics8060620_

Round 1

Reviewer 1 Report

This paper studies the effect of high-power amplifier on the generalized frequency division multiplexing scheme. The conclusion is based on the amplifier model and performance simulation results of GFDM based communications system. The point of experiments is clear and support the claims. However, some of the terms are used loosely which confuses the reader and hurts the readability of this article.

English needs improvement and text should be proof read.

Abstract: Give a reference for Saleh model in the abstract.

Line 79 : “Nevertheless, based on the central limit theorem and for large subcarrier values, the form of nonlinear distortion may be assumed to be statistically independent.”

This statement is not accurate. Central limit theorem does not imply statistical independence. Please see [1].

Line 91 : “closed-form probability density function (pdf) signal-to-noise ratio (SNR) “ This statement is not clear.

Line 387: “BER was measured by the Euclidean distance between the two closest signal points in the constellation diagram” You mean for theoretical curve? It is not clear.

It would be interesting if you could differentiate the effect of each nonlinear distortions on OFDM, MIMO, GFDM individually.

The method proposed is not compared against other methods available in the literature.

[1] Ash, Robert B., B. Robert, Catherine A. Doleans-Dade, and A. Catherine. Probability and measure theory. Academic Press, 2000.

Author Response

RESPONSE TO REVIEWER 1 COMMENTS

Point 1: This paper studies the effect of high-power amplifier on the generalized frequency division multiplexing scheme. The conclusion is based on the amplifier model and performance simulation results of GFDM based communications system. The point of experiments is clear and support the claims. However, some of the terms are used loosely which confuses the reader and hurts the readability of this article.

English needs improvement and text should be proof read.

Response 1: Thank you for your positive comments and your suggestion.

We have revised the paper and have proofread it according to the reviewer's recommendations to make it easier to read.

Point 2: Abstract: Give a reference for Saleh model in the abstract.

Response 2: Thank you so much for your comment.

We have inserted about the Saleh Model in the abstract on Line 23.

Point 3: Line 79: “Nevertheless, based on the central limit theorem and for large subcarrier values, the form of nonlinear distortion may be assumed to be statistically independent.”

This statement is not accurate. Central limit theorem does not imply statistical independence. Please see [1].

[1] Ash, Robert B., B. Robert, Catherine A. Doleans-Dade, and A. Catherine. Probability and measure theory. Academic Press, 2000.

Response 3: Thank you for your positive comments.

We have revised the sentence according to the understanding of the central limit theory that we read in the references given.

Point 4: Line 91: “closed-form probability density function (pdf) signal-to-noise ratio (SNR) “. This statement is not clear.

Response 4: Thank you for your positive comments.

We have revised it so that the sentence becomes clearer. “This study investigated the effect of HPA nonlinearity on MIMO-GFDM system and derived the new signal-to-noise (SNR) ratios in the scenario of nonlinear MIMO-GFDM system.”

Point 5: Line 387: “BER was measured by the Euclidean distance between the two closest signal points in the constellation diagram” You mean for theoretical curve? It is not clear.

Response 5: Thank you for your positive comments.

We have corrected the sentence to make it clearer. “Theoretical BER was calculated by the Euclidean distance between the two closest signal points in the constellation diagram which reduces the probability of errors”.

Point 6: It would be interesting if you could differentiate the effect of each nonlinear distortions on OFDM, MIMO, GFDM individually.

Response 6: Thank you for your suggestion.

We have inserted the results of the comparison of the effects of nonlinear distortion between OFDM and GFDM on line 411-416. Please see the references numbers 12 and 32.

[12] Jayati, A.E.; Wirawan; Suryani, T.; Endroyono. Characteristic of HPA Nonlinear Distortion Effects in MIMO GFDM Systems, in 2018 ICTC, Jeju Island, October 17-19. 2018, pp. 379-384.

[32] Jayati, A.E.; Wirawan; Suryani, T. Analysis of Non-Linear Distortion Effect Model Based on Saleh in GFDM System, in IEEE International Conference on Comnetsat , Semarang, October 6 to 7, 2017, pp. 13-18.

Point 7: The method proposed is not compared against other methods available in the literature.

Response 7: Thank you for your comments and suggestion.

The proposed method has been compared with OFDM, shown in Figure 14. In the next work, we will add a comparison with other methods, for example, OFDM, not only on BER analysis.

Reviewer 2 Report

Reviewer’s Recommendation:

This candidate paper needs major revisions.

Summary

This work is relevant to the research of nonlinear distortion in MIMO GFDM systems

General comments

The authors generally worked towards investigating a case study relevant to nonlinear distortion.  They have tried to show the benefits of their methods but somewhere in the way the reader loses interest as there is the feeling all these are already known. The innovation should be emphasized more inside text along with accomplishing the promising goals. The authors state many goals at the beginning while at eh conclusion they only speak for the importance of predistortion. This presentation needs work in order to be aacptable. Moreover there are also problems in garammar and expression. Please ask for help from a native speaker.

Suggested Improvements

Apart from the aforementioned, please see the following suggestions. Please give the appropriate attention and revise accordingly:

1.       You mention various contributions of this paper in the ABSTRACT. It could be better to and classify them in order to report as (1), (2), etc. whereas a large number should definitely strengthen this presentation. 

2.       You mention that "GFDM is able to overcome the weakness of OFDM, which is a high Peak Average Power Ratio (PAPR) value due to the use of fewer subcarriers in GFDM". This means that then this FDM is not so immune to channel fading as the signal will be transmitted through less subcarriers and consequently a significant number of lost subcarriers could terminate the QoS of the link. Please explain because a reader- specialist relevant to FDM will definitely criticize this.

3.       Relevant to OFDM and PAPR please see:

-          GFDM-Generalized Frequency Division Multiplexing. In VTC Spring (pp. 1-4).

-          Preliminary BER Study of a TC-OFDM system operating under noisy conditions Journal of Engineering Science and Technology Review Vol. 9:  No. 4.  pp. 13-16 Oct. 2016.

-          Enhanced PAPR in OFDM without deteriorating BER performance. International Journal of Communications, Network and System Sciences, 4(03), 164.

4.       You mention that "GFDM as a candidate waveforms for 5G also still uses high M-ary modulation". Please revise expression.

5.       Relevant to MIMO please see:

System performance of an LTE MIMO downlink in various fading environments. In Proceedings of the International Conference on Ambient Media and Systems, Porto, Portugal, 24–25 March 2011; pp. 36–43.

6.       Reference [18] is followed by reference [20]. Please revise.

7.       You mention that "The results are supported by Matlab simulation of the MIMO GFDM system in the AWGN channel". What about Ricean and Rayleigh fading?

8.       You mention that "The m subcarriers are up sampled by factor N to be converted into an impulse signal sequence". Please explain.

9.       Figure 1 should be of higher analysis.

10.   You mention that "and Lp is the length of the prototype filter". Where is the prototype filter?

11.   What specifications do you use for adding CP? Why not even Zero padding? Please see: In depth analysis of noise effects in orthogonal frequency division multiplexing systems, utilising a large number of subcarriers. In AIP Conference Proceedings (Vol. 1203, No. 1, pp. 967-972). AIP.

12.   Please explain the derivation of equation (6) from (5).

13.   Relevant to predistortion, please mention about the used simulink model or the utilized code.

14.   You mention that "The signal received for each antenna ymNL(n) was replaced by a superposition of an independent fading signal". Please identify fading signal and explain in details.

15.   You mention that "Hm.k(n) is a circular time domain channel matrix". Please explain why you chose that. It is essential.

16.   Relevant to Table "Algorithm Modelling of Predistorter for MIMO-GFDM Systems", please rewrite in a more acceptable form.

17.   You mention that "The AM/PM characteristics indicate the phase distortion was in a maximum phase shift of 40 degrees". This shows that all the channel was drifted by 40 degrees due to Rayleigh probably. How? Please explain.

18.   In Table 1 relevant to various parameters, these should have a range of values. E.g The Roll-of-factor and the subcarriers.

19.   You mention that "Therefore, the predistorter capable of linearizing HPA becomes linear". Please explain stricter.

20.   You mention that "A high PAPR indicates a very linear PA is needed at the transmitter". This is not the case. High PAPR indicates high fluctuation due to orthogonality and indicates that a technique of reducing PAPR should be used.

21.   You mention that "However, moving the PA over the linear region produces nonlinear distortion effects". Please revise.

22.   Relevant to figure 12 there is definitely a phase offset that in this case should be correlated to BER in order to find the performance of the system in this case.

23.   In Figure 14 the needed signal to noise ratio is relevantly big. Have you considered to use a coding technique such as trellis, convolutional or turbo coding etc in order to decrease the needed power?

24.   Conclusions are not properly fulfilled. Please revise.

Author Response

RESPONSE TO REVIEWER 2 COMMENTS

Point 1: The authors generally worked towards investigating a case study relevant to nonlinear distortion.  They have tried to show the benefits of their methods but somewhere in the way the reader loses interest as there is the feeling all these are already known. The innovation should be emphasized more inside text along with accomplishing the promising goals. The authors state many goals at the beginning while at eh conclusion they only speak for the importance of predistortion. This presentation needs work in order to be aacptable. Moreover there are also problems in garammar and expression. Please ask for help from a native speaker.

Response 1: Thank you for your positive comments and your suggestion.

We have investigated the problem of nonlinear distortion in the MIMO-GFDM system. We propose the application of predistorter methods to overcome the effects of nonlinear distortion on the system. This method, though it's been a long time, is a simple and effective method for overcoming with nonlinear distortion due to HPA. To the best of our knowledge, no research has investigated the application of this method to the MIMO-GFDM system. GFDM is a non-orthogonal waveform that is one of the candidates for the upcoming wireless waveform system. This is very interesting for us. We have corrected according to the reviewer's recommendations and have done proofread.

Point 2: You mention various contributions of this paper in the ABSTRACT. It could be better to and classify them in order to report as (1), (2), etc. whereas a large number should definitely strengthen this presentation.

Response 2: Thank you for your comments.

I have revised the abstract from line 20 to line 28.

The contribution of this paper is first, we investigate the effects of nonlinear distortion on the MIMO-GFDM system. We also derive the new closed-form expression Bit Error Rate (BER) in MIMO-GFDM system that uses a memoryless HPA, which is modeled with Saleh Model when passed in Additive White Gaussian Noise (AWGN) channel. This model was chosen because it is simple and has AM / AM and AM / PM curves. Second, we propose the application of techniques for linearization of each HPA predistorter on the transmitter side of the GFDM MIMO system separately. This predistorter is able to compensate for nonlinear distortion caused by HPA without memory operating in the saturation region. Finally, the main contribution of this paper is the statement that processing on the transmitter can reduce the effect of nonlinear distortion.

Point 3: You mention that "GFDM is able to overcome the weakness of OFDM, which is a high Peak Average Power Ratio (PAPR) value due to the use of fewer subcarriers in GFDM". This means that then this FDM is not so immune to channel fading as the signal will be transmitted through less subcarriers and consequently a significant number of lost subcarriers could terminate the QoS of the link. Please explain because a reader- specialist relevant to FDM will definitely criticize this.

Response 3: Thank you for your positive comments.

We have revised the sentence on line 36-42. The following is an explanation of the reasons GFDM can overcome OFDM weaknesses.

GFDM is able to overcome OFDM weaknesses, which is a high Peak to Average Power Ratio (PAPR) because OFDM signals are the sum of exponential terms limited to rectangular windows, while GFDM is the sum of limited exponential terms with Raised Cosine (RC) filters. Given the assumption of having a normalized filter, the average power of the GFDM signal is certainly reduced by applying windowing while remaining constant for OFDM. Therefore, PAPR is the same for the sum of exponential terms and OFDM signals, while statistically growing for GFDM [1].

GFDM reduces PAPR because the number of symbols M (M>1) in each subcarrier K. GFDM is a block-based multicarrier scheme that reduces OOB and PAPR radiation compared to OFDM, thus making GFDM future potential OFDM for 5G services such as Cognitive radio, Bitpipe communication, tactile internet, and so on. The pulse formation filter used in GFDM allows GFDM to achieve the desired performance metric [2].

[1] Sharifian, Z; Omidi, M.J; Farhang, A; Saeedi-Sourck, H. Polynomial-based compressing and iterative expanding for PAPR reduction in GFDM. ICEE 2015 - Proc 23rd Iran Conf Electr Eng. 2015, 518-523.

[2] Antapurkar, S.K; Pandey, A; Gupta, K.K. GFDM performance in terms of BER, PAPR and OOB and comparison to OFDM system. AIP Conf Proc. 2016, 1715.

Point 4: Relevant to OFDM and PAPR please see:

-        GFDM-Generalized Frequency Division Multiplexing. In VTC Spring (pp. 1-4).

-         Preliminary BER Study of a TC-OFDM system operating under noisy conditions Journal of Engineering Science and Technology Review Vol. 9:  No. 4.  pp. 13-16 Oct. 2016.

-          Enhanced PAPR in OFDM without deteriorating BER performance. International Journal of Communications, Network and System Sciences, 4(03), 164.

Response 4: Thank you for your suggestion.

We have read the literature and added it to the bibliography.

Research on PAPR in OFDM has been carried out [3-5].

Point 5: You mention that "GFDM as a candidate waveforms for 5G also still uses high M-ary modulation". Please revise expression.

Response 5: Thank you for your comment.

We have improved these expressions so that the reader will understand more easily.

“New waveforms for the physical layer (PHY) and diverse requirements for 5G cellular systems are being studied, where one of the candidates is non-orthogonal Generalized Frequency Division Multiplexing (GFDM). This new wave still uses high M-ary modulation”.

Point 6: Relevant to MIMO please see:

System performance of an LTE MIMO downlink in various fading environments. In Proceedings of the International Conference on Ambient Media and Systems, Porto, Portugal, 24–25 March 2011; pp. 36–43.

Response 6: Thank you for your suggestion reference.

This paper discusses the performance of the LTE MIMO downlink system in various fading environments. This reference was added to line 78.

Research that discusses the performance of LTE MIMO downlink systems in various fading environments has been carried out [21]”.

Point 7: Reference [18] is followed by reference [20]. Please revise.

Response 7: Thank you for your comments.

We have revised on line 81.

Point 8: You mention that "The results are supported by Matlab simulation of the MIMO GFDM system in the AWGN channel". What about Ricean and Rayleigh fading?

Response 8: Thank you for your comments and suggestion.

In this paper, only the Matlab simulation results from the GFDM MIMO system on the AWGN channel are displayed. In the next work, the performance of the MIMO-GFDM system on the Rayleigh channel and Rician Fading will be examined.

Point 9: You mention that "The m subcarriers are up sampled by factor N to be converted into an impulse signal sequence". Please explain.

Response 9: Thank you for your comments.

We have corrected the sentence on line 125.  “Each sub-symbol results from the decomposition, upsampled by factor ? to be converted into an impulse signal sequence”.

Point 10: Figure 1 should be of higher analysis.

Response 10: Thank you for your suggestions.

We have added a more detailed explanation from Figure 1. Changes can be seen in lines 134-139.

 is a prototype filter g (n) with the shifted version in time and frequency, k is the subcarrier index and m is for the subsymbol index. The prototype filter above consists of two components, namely and complex exponential numbers. The first component is a circular filter that functions as a time slider to distinguish between one symbol and the symbol others based on the timeslot. While the second component functions as a signal shifter in the frequency domain.

Point 11: You mention that "and Lp is the length of the prototype filter". Where is the prototype filter?

Response 11: Thank you for your comment.

We have added an explanation of the prototype filter on line 134-139.

Point 12: What specifications do you use for adding CP? Why not even Zero padding? Please see: In depth analysis of noise effects in orthogonal frequency division multiplexing systems, utilising a large number of subcarriers. In AIP Conference Proceedings (Vol. 1203, No. 1, pp. 967-972). AIP.

Response 12: Thank you for your comments and suggestion reference.

The cyclic prefix is added to GFDM to maintain the circular structure of the transmission signal and to allow the frequency domain even distribution to the receiver after a multi-lane effect is applied in the channel[1]. The insertion of a cyclic prefix in each block GFDM instead of in each symbol increases the spectral efficiency[2]. In the MIMO-GFDM system, CP is taken from the final copy of each the GFDM symbol is then placed at the beginning of the frame. The number of symbols in this study is KxM which is 120, so the number of CPs is used is 0.25xK = 0.25x8 = 2.

[1] Michailow N, Krone S, Lentmaier M, Fettweis G. Bit Error Rate Performance of Generalized Frequency Division Multiplexing. 2012 IEEE Veh Technol Conf (VTC Fall). 2012:1-5. doi:10.1109/VTCFall.2012.6399305.

[2] Matthe M, Michailow N, Gaspar I, Fettweis G. Influence of pulse shaping on bit error rate performance and out of band radiation of Generalized Frequency Division Multiplexing. 2014 IEEE Int Conf Commun Work ICC 2014. 2014:43-48. doi:10.1109/ICCW.2014.6881170.

Point 13: Please explain the derivation of equation (6) from (5).

Response 13: Thank you for your suggestion.

[31] proposed the functions  and  with two-parameter formulas such as equations 5 and 6. Where is an odd function of r, which represents the AM-to-AM conversion, and is a function even number of r, which represents the AM-to-PM conversion. If r is very large r, is proportional to , and approaches constant.

Point 14: Relevant to predistortion, please mention about the used simulink model or the utilized code.

Response 14: Thank you for your suggestion.

MIMO-GFDM system simulation with predistorters using matlab according to equation (10).

%Predistorter

xpred = abs(nx); phi = angle(nx); 

af=2.1587; bf=1.1517; ag=4.0033; bg=9.104;

c = (af^2)/(4*bf); Lim = sqrt(c); r = zeros(1,length(xpred)); psi0 = [];

for i = 1:1:length(xpred)

     if xpred(1,i) > 0 && xpred(1,i) < c

         r(1,i) = (af-sqrt((af^2)-4*bf*xpred(1,i)^2))./(2*bf*xpred(1,i));

     elseif xpred(1,i) >= c

         r(1,i) = 1/sqrt(bf);

     end   

     psi0 = [psi0 (ag*r(1,i)^2)/(1+bg*r(1,i)^2)];

 end

 theta = phi - psi0;

 rr=r.*exp(1i*theta);

Point 15: You mention that "The signal received for each antenna ymNL(n) was replaced by a superposition of an independent fading signal". Please identify fading signal and explain in details.

Response 15: Thank you for your comment.

We have revised the sentence as shown in line 240. In this study, we use the AWGN channel, for the Rayleigh and the Rician channel will be our next study.

“The signal received at each yNL antenna is formed by the superposition of the signal independently. This is related to the Nt antenna which shares the same space frequency resources”.

Point 16: You mention that "Hm.k(n) is a circular time domain channel matrix". Please explain why you chose that. It is essential.

Response 16: Thank you for your comment.

 is a circular time domain channel matrix. This is because the application of the cyclic prefix before transmission, the channel matrix becomes a circular matrix, the decomposition of the eigenvectors leads to diagonalization of the channel matrix [1]. While on the recipient side using time domain equalization techniques that offer better performance compared to frequency [2].

[1]Baig S. Frequency Domain Channel Equalization Using Circulant Channel Matrix Diagonalization. 4.

[2]R. Kumar, S. Malarvizhi and S. Jayashri, 2008. Time-Domain Equalization Technique for Intercarrier Interference Suppression in OFDM Systems. Information Technology Journal, 7: 149-154.

Point 17: Relevant to Table "Algorithm Modelling of Predistorter for MIMO-GFDM Systems", please rewrite in a more acceptable form.

Response 17: Thank you for your comment.

We have revised the table so it's easy to read.

Point 18: You mention that "The AM/PM characteristics indicate the phase distortion was in a maximum phase shift of 40 degrees". This shows that all the channel was drifted by 40 degrees due to Rayleigh probably. How? Please explain.

Response 18: Thank you for your comment.

We have corrected the sentence on line 288.

“AM / PM characteristics have been normalized to produce a zero degree phase shift at -10 dBm”.

Point 19: In Table 1 relevant to various parameters, these should have a range of values. E.g The Roll-of-factor and the subcarriers.

Response 19: Thank you for your comment.

The range of values of roll of factors is from 0 to 1. While the number of sub-carriers is a multiple of 2.

In this study, we used the value of a roll of factor 0.3 and the number of subcarriers 8.

Point 20: You mention that "Therefore, the predistorter capable of linearizing HPA becomes linear". Please explain stricter.

Response 20: Thank you for your comment.

The aim of the predistorter is to modify the signal on the transmitter so that the nonlinear distortion seen in the PA output decreases. Predistorters can be implemented as inverses directly from the characteristics of the HPA[1].

[1] Yoffe, I.; Wulich, D. Predistorter for MIMO System with Nonlinear Power Amplifiers. IEEE Trans. On Commss, 2017, 65, 8, 3288-3301.

Point 21: You mention that "A high PAPR indicates a very linear PA is needed at the transmitter". This is not the case. High PAPR indicates high fluctuation due to orthogonality and indicates that a technique of reducing PAPR should be used.

Response 21: Thank you for your comments and suggestion.

We have deleted the sentence because it is not in accordance with the purpose of this paper.

Point 22: You mention that "However, moving the PA over the linear region produces nonlinear distortion effects". Please revise.

Response 22: Thank you for your comments and suggestion.

We have revised the sentence on line 359-362.

PA will be more power efficient if it operates close to the saturation region. Similarly, if the PA gets closer to the saturation region, the amount of nonlinear distortion is greater, which will reduce system performance.

Point 23: Relevant to figure 12 there is definitely a phase offset that in this case should be correlated to BER in order to find the performance of the system in this case.

Response 23: Thank you for your comments and suggestion.

Phase distortion occurs when an amplifier does not delay all frequency components by the same amount. The output phase distortion has a logarithmic shape instead of being linearly increasing as in the Saleh model. This means that the output phase shift is almost constant at high input amplitude values[1]. Phase distortion causes the constellation of signals on the receiving side no longer at the original point. This will affect the decision point area on the receiving side. As a result, the signal received will be more difficult to detect correctly or can be detected but an error signal is received information resulting in a decrease in performance.

[1] Jantunen, P. Modeling of Nonlinear Power Amplifiers for Wireless Communications. 2004

Point 24: In Figure 14 the needed signal to noise ratio is relevantly big. Have you considered to use a coding technique such as trellis, convolutional or turbo coding etc in order to decrease the needed power?

Response 24: Thank you for your comments and suggestion.

In the next work Alamouti Space Time Coding will be added to the MIMO-GFDM system.

Point 25: Conclusions are not properly fulfilled. Please revise.

Response 25: Thank you for your suggestion.

We have corrected the conclusions to fit the research objectives in this paper.

“This study investigated and analyzed the effect of the HPA nonlinear distortion GFDM MIMO on the shape of the spectrum, the PAPR and BER in AWGN channel. This research has derived the closed-form signal-to-noise ratio (SNR) in a nonlinear MIMO-GFDM system scenario. Furthermore, it applied predistorter techniques to mitigate distortion effects of the HPA nonlinear MIMO GFDM on the sender side.

The effectiveness of the proposed predistorter was illustrated by the simulation results which showed the performance comparison of MIMO GFDM with the HPA without predistorter. The result of MIMO GFDM using predistorter mitigated the effects of HPA nonlinear distortion. The system with predistorter was shown by a better spectrum parameters, good PAPR and lower BER values. The predistorter compensated for serious nonlinear distortion caused by the HPA memory which operated in the saturation region. It is clear that the transmitter processing method reduces the effect of nonlinear distortion.

 This study only overcame nonlinear distortions in MIMO-GFDM system transmitters. In MIMO systems, interference is not only nonlinear distortion but also crosstalk due to interference between different signal paths. Therefore, the compensation technique of the joint effect of nonlinear distortion and crosstalk in MIMO-GFDM systems should be considered in future studies.

Round 2

Reviewer 1 Report

This paper needs more work. 

This paper is full of vague statement which has not been properly addressed, for example "The main contribution of this paper is the statement that the processing at the transmitter may reduce the effect of nonlinear distortion." You can find multiple other examples.

Saleh model should be cited properly in the abstract. 

"The contribution of this paper is, first, ..." Please use a professional editor to correct the use of english in this paper.

Author Response

Point 1: This paper needs more work. 

Response 1: Thank you for your positive comments and your suggestion.

We have improved this paper according to the comments of the reviewers. For some suggestions from reviewers, it will be a plan for our next research.

1.      In the next work Alamouti Space Time Coding will be added to the MIMO-GFDM system.

2.      This study only overcame nonlinear distortions in MIMO-GFDM system transmitters. In MIMO systems, interference is not only nonlinear distortion, but also crosstalk due to the interference between different signal paths. Therefore, the compensation technique of the joint effect of nonlinear distortion and crosstalk in MIMO-GFDM systems should be considered in future studies.

3.      The predistorters linearize nonlinear distortions successful due to HPA for larger IBO. Future studies should focus on small IBO values to have a smaller BER value.

Point 2: This paper is full of vague statement which has not been properly addressed, for example "The main contribution of this paper is the statement that the processing at the transmitter may reduce the effect of nonlinear distortion." You can find multiple other examples.

Response 2: Thank you so much for your comment.

We have corrected the unclear statement.

“The main contribution of this paper is to investigate the predistorter which can linearize nonlinear distortion in MIMO-GFDM transmitters. The performance of the proposed scheme is evaluated in terms of spectrum analysis, PAPR analysis, constellation diagram, and BER analysis. The simulation results show that the proposed predistorter design succeed in compensating nonlinear distortions caused by HPA for large IBO values.”

Point 3: Saleh model should be cited properly in the abstract.

Response 3: Thank you for your positive comments.

We also derive the new closed-form expression Bit Error Rate (BER) in MIMO-GFDM system that uses a memoryless HPA, which is modeled by using Saleh Model [32] when passed in the Additive White Gaussian Noise (AWGN) channel.

Point 4: "The contribution of this paper is, first, ..." Please use a professional editor to correct the use of english in this paper.

Response 4: Thank you for your positive comments.

We have proofread this paper.

Contributions to this paper can be divided into three. Firstly, we investigate the effects of nonlinear distortion on the MIMO-GFDM system. We also derive the new closed-form expression Bit Error Rate (BER) in MIMO-GFDM system that uses a memoryless HPA, which is modeled by using Saleh Model when passed in the Additive White Gaussian Noise (AWGN) channel. This model was chosen because it is simple and has AM/AM and AM/PM curves. Secondly, we propose the application of techniques for linearization of each HPA predistorter on the transmitter side of the GFDM MIMO system separately.

Reviewer 2 Report

The authors have made the appropriate revisions but still some minors should be conducted. No new review is needed from my part if possible. Please see:

1.       Please revise expression "If r is very large r...".

2.       Please import Response 14 inside manuscript or else declare that the code is available upon request (Please discuss it with Editor).

3.       Please incorporate Response 16 inside manuscript with a more appropriate grammatical expression.

4.       Line 269: Please revert "phasa predistorter" to "phase of predistorter".

5.       Relevant to Response 19 please take into consideration that you have to justify why you chose roll of factor and specific number of subcarriers.

6.       Please incorporate Response 24 probably as a future scope.

Author Response

Point 1: Please revise expression "If r is very large r...".

Response 1: Thank you for your positive comments and your suggestion.

We have corrected the sentence. 

“If r is very large, A[r(n)] is proportional to 1/r, and Φ[r(n)] approaches constant”

Point 2: Please import Response 14 inside manuscript or else declare that the code is available upon request (Please discuss it with Editor).

Response 2: Thank you for your comments.

We will include the program code response 14 in supplement material.

Point 3: Please incorporate Response 16 inside manuscript with a more appropriate grammatical expression.

Response 3: Thank you for your positive comments.

H_(m,k) (n) is a circular time domain channel matrix. The channel matrix becomes a circular matrix because of the application of a cyclic prefix before transmission. In addition, the decomposition of the eigenvectors leads to the diagonalization of the channel matrix [1]. While at the receiver side using time domain equalization techniques that offer better performance compared to frequency domain equalization techniques [2].

[1]Baig, S; Rehman, Fazal-ur. Frequency Domain Channel Equalization Using Circulant Channel Matrix Diagonalization. In 2005 Pakistan Section Multitopic Conference. 2005, pp 1-5.

[2]Kumar, R; Malarvizhi, S;Jayashri, S. Time-Domain Equalization Technique for Intercarrier Interference Suppression in OFDM Systems. Information Technology Journal, 2008, vol 7, pp.149-154.

Point 4: Line 269: Please revert "phasa predistorter" to "phase of predistorter".

Response 4: Thank you for your suggestion.

We have corrected it.

Point 5: Relevant to Response 19 please take into consideration that you have to justify why you chose roll of factor and specific number of subcarriers.

Response 5: Thank you for your comment.

The selection of a roll-of-factor value between 0 and 1 because Roll-of-factor (α) which has a value of 0 indicates the use of a rectangular pulse in the pulse shaping process. Whereas α which has a value of 1 indicates the use of a non-rectangular pulse. Increasing the roll-off factor increases the SNR gap [1].

Selection of the number of subcarriers 8, because this value is the smallest value used to compare the performance between GFDM and OFDM. The fragmentation spectrum which greatly increases the spectrum efficiency. [2]

[1] Matthe M, Michailow N, Gaspar I, Fettweis G. Influence of pulse shaping on bit error rate performance and out of band radiation of Generalized Frequency Division Multiplexing. 2014 IEEE Int Conf Commun Work ICC 2014. 2014:43-48. doi:10.1109/ICCW.2014.6881170.

[2] Wu J, Ma X, Qi X, Babar Z, Zheng W. Influence of pulse shaping filters on PAPR performance of underwater 5G communication system technique: GFDM. Wirel Commun Mob Comput. 2017;2017. doi:10.1155/2017/4361589.

Point 6: Please incorporate Response 24 probably as a future scope 

Response 6: Thank you for your suggestion reference.

We have added to the next work plan.

In the next work Alamouti Space Time Coding will be added to the MIMO-GFDM system.
